# Relaxed Quantization for Discretized Neural Networks

**Christos Louizos**[*]
University of Amsterdam
TNO Intelligent Imaging
c.louizos@uva.nl

**Matthias Reisser**
QUVA Lab
University of Amsterdam
m.reisser@uva.nl

**Tijmen Blankevoort**
Qualcomm AI Research
tijmen@qti.qualcomm.com

**Efstratios Gavves**
QUVA Lab
University of Amsterdam
egavves@uva.nl

**Max Welling**
University of Amsterdam
Qualcomm
m.welling@uva.nl

## Abstract

Neural network quantization has become an important research area due to its great impact on deployment of large models on resource constrained devices. In order to train networks that can be effectively discretized without loss of performance, we introduce a differentiable quantization procedure. Differentiability can be achieved by transforming continuous distributions over the weights and activations of the network to categorical distributions over the quantization grid. These are subsequently relaxed to continuous surrogates that can allow for efficient gradient-based optimization. We further show that stochastic rounding can be seen as a special case of the proposed approach and that under this formulation the quantization grid itself can also be optimized with gradient descent. We experimentally validate the performance of our method on MNIST, CIFAR 10 and Imagenet classification.

## 1 Introduction

Neural networks excel in a variety of large scale problems due to their highly flexible parametric nature. However, deploying big models on resource constrained devices, such as mobile phones, drones or IoT devices is still challenging because they require a large amount of power, memory and computation. Neural network compression is a means to tackle this issue and has therefore become an important research topic.

Neural network compression can be, roughly, divided into two not mutually exclusive categories: pruning and quantization. While pruning (LeCun et al., 1990; Han et al., 2015) aims to make the model "smaller" by altering the architecture, quantization aims to reduce the precision of the arithmetic operations in the network. In this paper we focus on the latter. Most network quantization methods either simulate or enforce discretization of the network during training, e.g. via rounding of the weights and activations. Although seemingly straighforward, the discontinuity of the discretization makes the gradient-based optimization infeasible. The reason is that there is no gradient of the loss with respect to the parameters. A workaround to the discontinuity are the "pseudo-gradients" according to the straight-through estimator (Bengio et al., 2013), which have been successfully used for training low-bit width architectures at e.g. Hubara et al. (2016); Zhu et al. (2016).

The purpose of this work is to introduce a novel quantization procedure, Relaxed Quantization (RQ). RQ can bypass the non-differentiability of the quantization operation during training by smoothing it appropriately. The contributions of this paper are four-fold: **First**, we show how to make the set of quantization targets part of the training process such that we can optimize them with gradient descent. **Second**, we introduce a way to discretize the network by converting distributions over the weights and activations to categorical distributions over the quantization grid. **Third**, we show that we can obtain a "smooth" quantization procedure by replacing the categorical distributions with

---

[*]Work done while interning at Qualcomm AI Research.

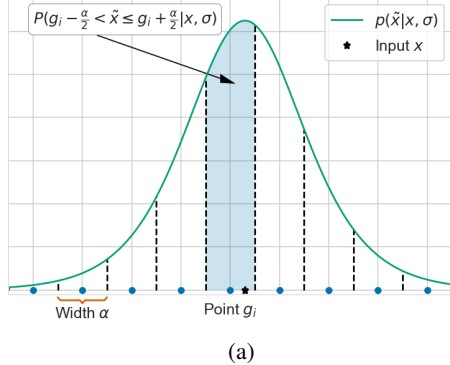 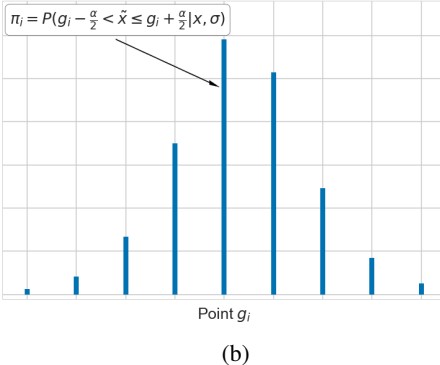

(a)  (b)

Figure 1: The proposed discretization process. **(a)** Given a distribution $p(\tilde{x})$ over the real line we partition it into $K$ intervals of width $\alpha$ where the center of each of the intervals is a grid point $g_i$. The shaded area corresponds to the probability of $\tilde{x}$ falling inside the interval containing that specific $g_i$. **(b)** Categorical distribution over the grid obtained after discretization. The probability of each of the grid points $g_i$ is equal to the probability of $\tilde{x}$ falling inside their respective intervals.

concrete (Maddison et al., 2016; Jang et al., 2016) equivalents. **Finally** we show that stochastic rounding (Gupta et al., 2015), one of the most popular quantization techniques, can be seen as a special case of the proposed framework. We present the details of our approach in Section 2, discuss related work in Section 3 and experimentally validate it in Section 4. Finally we conclude and provide fruitful directions for future research in Section 5.

## 2 RELAXED QUANTIZATION FOR DISCRETIZING NEURAL NETWORKS

The central element for the discretization of weights and activations of a neural network is a quantizer $q(\cdot)$. The quantizer receives a (usually) continous signal as input and discretizes it to a countable set of values. This process is inherently lossy and non-invertible: given the output of the quantizer, it is impossible to determine the exact value of the input. One of the simplest quantizers is the rounding function:

$$q(x) = \alpha \left\lfloor \frac{x}{\alpha} + \frac{1}{2} \right\rfloor,$$

where $\alpha$ corresponds to the step size of the quantizer. With $\alpha = 1$, the quantizer rounds $x$ to its nearest integer number.

Unfortunately, we cannot simply apply the rounding quantizer to discretize the weights and activations of a neural network. Because of the quantizers' lossy and non-invertible nature, important information might be destroyed and lead to a decrease in accuracy. To this end, it is preferable to train the neural network while simulating the effects of quantization during the training procedure. This encourages the weights and activations to be robust to quantization and therefore decreases the performance gap between a full-precision neural network and its discretized version.

However, the aforementioned rounding process is non-differentiable. As a result, we cannot directly optimize the discretized network with stochastic gradient descent, the workhorse of neural network optimization. In this work, we posit a "smooth" quantizer as a possible way for enabling gradient based optimization.

### 2.1 LEARNING (FIXED POINT) QUANTIZERS VIA GRADIENT DESCENT

The proposed quantizer comprises four elements: a vocabulary, its noise model and the resulting discretization procedure, as well as a final relaxation step to enable gradient based optimization.

**The first element** of the quantizer is the vocabulary: it is the set of (countable) output values that the quantizer can produce. In our case, this vocabulary has an inherent structure, as it is a grid of ordered

scalars. For fixed point quantization the grid $\mathcal{G}$ is defined as

$$\mathcal{G} = \left[ -2^{b-1}, \ldots, 0, \ldots, 2^{b-1} - 1 \right], \tag{1}$$

where $b$ is the number of available bits that allow for $K = 2^b$ possible integer values. By construction this grid of values is agnostic to the input signal $x$ and hence suboptimal; to allow for the grid to adapt to $x$ we introduce two free parameters, a scale $\alpha$ and an offset $\beta$. This leads to a learnable grid via $\hat{\mathcal{G}} = \alpha\mathcal{G} + \beta$ that can adapt to the range and location of the input signal.

**The second element** of the quantizer is the assumption about the input noise $\epsilon$; it determines how probable it is for a specific value of the input signal to move to each grid point. Adding noise to $x$ will result in a quantizer that is, on average, a smooth function of its input. In essense, this is an application of variational optimization (Staines & Barber, 2012) to the non-differentiable rounding function, which enables us to do gradient based optimization.

We model this form of noise as acting additively to the input signal $x$ and being governed by a distribution $p(\epsilon)$. This process induces a distribution $p(\tilde{x})$ where $\tilde{x} = x + \epsilon$. In the next step of the quantization procedure, we discretize $p(\tilde{x})$ according to the quantization grid $\hat{\mathcal{G}}$; this neccesitates the evaluation of the cumulative distribution function (CDF). For this reason, we will assume that the noise is distributed according to a zero mean logistic distribution with a standard deviation $\sigma$, i.e. $L(0, \sigma)$, hence leading to $p(\tilde{x}) = L(x, \sigma)$. The CDF of the logistic distribution is the sigmoid function which is easy to evaluate and backpropagate through. Using Gaussian distributions proved to be less effective in preliminary experiments. Other distributions are conceivable and we will briefly discuss the choice of a uniform distribution in Section 2.3.

**The third element** is, given the aforementioned assumptions, how the quantizer determines an appropriate assignment for each realization of the input signal $x$. Due to the stochastic nature of $\tilde{x}$, a deterministic round-to-nearest operation will result in a stochastic quantizer for $x$. Quantizing $x$ in this manner corresponds to discretizing $p(\tilde{x})$ onto $\hat{\mathcal{G}}$ and then sampling grid points $g_i$ from it. More specifically, we construct a categorical distribution over the grid by adopting intervals of width equal to $\alpha$ centered at each of the grid points. The probability of selecting that particular grid point will now be equal to the probability of $\tilde{x}$ falling inside those intervals:

$$p(\hat{x} = g_i | x, \sigma) = P(\tilde{x} \leq (g_i + \alpha/2)) - P(\tilde{x} < (g_i - \alpha/2))) \tag{2}$$
$$= \text{Sigmoid}((g_i + \alpha/2 - x)/\sigma) - \text{Sigmoid}((g_i - \alpha/2 - x)/\sigma), \tag{3}$$

where $\hat{x}$ corresponds to the quantized variable, $P(\cdot)$ corresponds to the CDF and the step from Equation 2 to Equation 3 is due to the logistic noise assumption. A visualization of the aforementioned process can be seen in Figure 1. For the first and last grid point we will assume that they reside within $(g_0 - \alpha/2, g_0 + \alpha/2]$ and $(g_K - \alpha/2, g_K + \alpha/2]$ respectively. Under this assumption we will have to truncate $p(\tilde{x})$ such that it only has support within $(g_0 - \alpha/2, g_K + \alpha/2]$. Fortunately this is easy to do, as it corresponds to just a simple modification of the CDF:

$$P(\tilde{x} \leq c | \tilde{x} \in (g_0 - \alpha/2, g_K + \alpha/2]) = \frac{P(\tilde{x} \leq c) - P(\tilde{x} < (g_0 - \alpha/2))}{P(\tilde{x} \leq (g_K + \alpha/2)) - P(\tilde{x} < (g_0 - \alpha/2))}. \tag{4}$$

Armed with this categorical distribution over the grid, the quantizer proceeds to assign a specific grid value to $\hat{x}$ by drawing a random sample. This procedure emulates quantization noise, which prevents the model from fitting the data. This noise can be reduced in two ways: by clustering the weights and activations around the points of the grid and by reducing the logistic noise $\sigma$. As $\sigma \to 0$, the CDF converges towards the step function, prohibiting gradient flow. On the other hand, if $\epsilon$ is too high, the optimization procedure is very noisy, prohibiting convergence. For this reason, during optimization we initialize $\sigma$ in a sensible range, such that $L(x, \sigma)$ covers a significant portion of the grid. Please confer Appendix A for details. We then let $\sigma$ be freely optimized via gradient descent such that the loss is minimized. Both effects reduce the gap between the function that the neural network computes during training time vs. test time. We illustrate this in Figure 2.

**The fourth element** of the procedure is the relaxation of the non-differentiable categorical distribution sampling. While we can use an unbiased gradient estimator via REINFORCE (Williams, 1992), we opt for a continuous relaxation due to high variances with REINFORCE. This is achieved by replacing the categorical distribution with a concrete distribution (Maddison et al., 2016; Jang et al., 2016). This relaxation procedure corresponds to adopting a "smooth" categorical distribution that

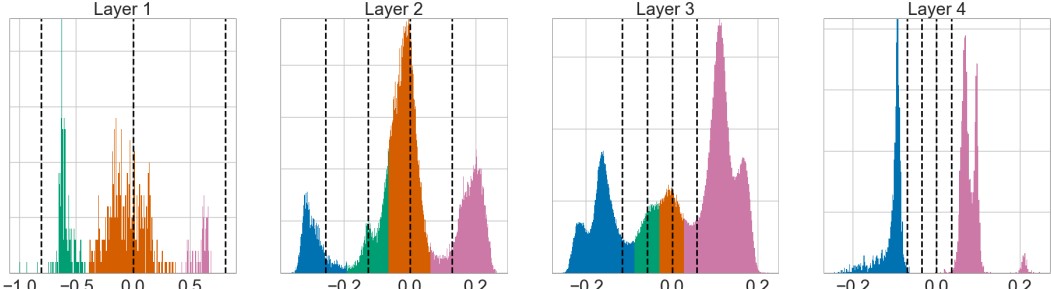

Figure 2: Best viewed in color. Illustration of the inductive bias obtained via training with the proposed quantizer; means of the logistic distribution over the weights for each layer of the LeNet-5 when trained with 2 bits per weight and activation. Each color corresponds to an assignment to a particular grid point and the vertical dashed lines correspond to the grid points ($\beta = 0$). We can clearly see that the real valued weights are naturally encouraged through training to cluster into multiple modes, one for each grid point. It should also be mentioned, that for the right and leftmost grid points the probability of selecting them is maximized by moving the corresponding weight furthest right or left respectively. Interestingly, we observe that the network converged to ternary weights for the input and (almost) binary weights for the output layer.

can be seen as a "noisy" softmax. Let $\pi_i$ be the categorical probability of sampling grid point $i$, i.e. $\pi_i = p(\hat{x} = g_i)$; the "smoothed" quantized value $\hat{x}$ can be obtained via:

$$u_i \sim \text{Gumbel}(0,1), \qquad z_i = \frac{\exp((\log \pi_i + u_i)/\lambda)}{\sum_j \exp((\log \pi_j + u_j)/\lambda)}, \qquad \hat{x} = \sum_{i=1}^{K} z_i g_i, \qquad (5)$$

where $z_i$ is the random sample from the concrete distribution and $\lambda$ is a temperature parameter that controls the degree of approximation, since as $\lambda \to 0$ the concrete distribution becomes a categorical.

We have thus defined a fully differentiable "soft" quantization procedure that allows for stochastic gradients for both the quantizer parameters $\alpha, \beta, \sigma$ as well as the input signal $x$ (e.g. the weights or the activations of a neural network). We refer to this algorithm as Relaxed Quantization (RQ). We summarize its forward pass as performed during training in Algorithm 1. It is also worthwhile to notice that if there were no noise at the input $x$ then the categorical distribution would have non-zero mass only at a single value, thus prohibiting gradient based optimization for $x$ and $\sigma$.

One drawback of this approach is that the smoothed quantized values defined in Equation 5 do not have to coincide with grid points, as $z$ is not a one-hot vector. Instead, these values can lie anywhere between the smallest and largest grid point, something which is impossible with e.g. stochastic rounding (Gupta et al., 2015). In order to make sure that only grid-points are sampled, we propose an alternative algorithm RQ ST in which we use the variant of the straight-through (ST) estimator proposed in Jang et al. (2016). Here we sample the actual categorical distribution during the forward pass but assume a sample from the concrete distribution for the backward pass. While this gradient estimator is obviously biased, in practice it works as the "gradients" seem to point towards a valid direction. This effect was also recently studied at Yin et al. (2019). We perform experiments with both variants.

After convergence, we can obtain a "hard" quantization procedure, i.e. select points from the grid, at test time by either reverting to a categorical distribution (instead of the continuous surrogate) or by rounding to the nearest grid point. In this paper we chose the latter as it is more aligned with the low-resource environments in which quantized models will be deployed. Furthermore, with this goal in mind, we employ two quantization grids with their own learnable scalar $\alpha, \sigma$ (and potentially $\beta$) parameters for each layer; one for the weights and one for the activations.

## 2.2 Scalable quantization via a local grid

Sampling $\hat{x}$ based on drawing $K$ random numbers for the concrete distribution as described in Equation 5 can be very expensive for larger values of $K$. Firstly, drawing $K$ random numbers

| **Algorithm 1** Quantization during training. | **Algorithm 2** Quantization during testing. |
|---|---|
| **Require:** Input $x$, grid $\hat{\mathcal{G}}$, scale of the grid $\alpha$, scale of noise $\sigma$, temperature $\lambda$, fuzz param. $\epsilon$ | **Require:** Input $x$, scale and offset of the grid $\alpha, \beta$, minimum and maximum values $g_0, g_K$ |
| $r = [\hat{\mathcal{G}} - \alpha/2, g_K + \alpha/2]$    # interval points | $y = \alpha \cdot \text{round}((x - \beta)/\alpha) + \beta$ |
| $c = \text{Sigmoid}((r - x)/\sigma)$    # evaluate CDF | **return** $\min(g_K, \max(g_0, y))$ |
| $\pi_i = \frac{c[i+1] - c[i] + \epsilon}{c[K+1] - c[1] + K\epsilon}$    # categorical distr. | |
| $z \sim \text{Concrete}(\pi, \lambda)$ | |
| **return** $\sum_i z_i g_i$ | |

for every individual weight and activation in a neural network drastically increases the number of operations required in the forward pass. Secondly, it also requires keeping many more numbers in memory for gradient computations during the backward pass. Compared to a standard neural network or stochastic rounding approaches, the proposed procedure can thus be infeasible for larger models and datasets.

Fortunately, we can make sampling $\hat{x}$ independent of the grid size by assuming zero probability for grid-points that lie far away from the signal $x$. Specifically, by only considering grid points that are within $\delta$ standard deviations away from $x$, we truncate $p(\tilde{x})$ such that it lies within a "localized" grid around $x$.

To simplify the computation required for determining the local grid elements, we choose the grid point closest to $x$, $\lfloor x \rceil$, as the center of the local grid (Figure 3). Since $\sigma$ is shared between all elements of the weight matrix or activation, the local grid has the same width for every element.

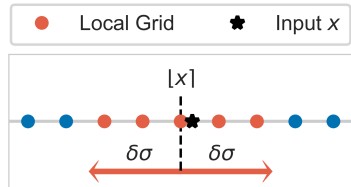

The computation of the probabilities over the localized grid is similar to the truncation happening in Equation 4 and the smoothed quantized value is obtained via a manner similar to Equation 5:

Figure 3: Local grid construction

$$P(\tilde{x} \le c | \tilde{x} \in (\lfloor x \rceil - \delta\sigma, \lfloor x \rceil + \delta\sigma]) = \frac{P(\tilde{x} \le c) - P(\tilde{x} < \lfloor x \rceil - \delta\sigma)}{P(\tilde{x} \le \lfloor x \rceil + \delta\sigma) - P(\tilde{x} < \lfloor x \rceil - \delta\sigma)} \tag{6}$$

$$\hat{x} = \sum_{g_i \in (\lfloor x \rceil - \delta\sigma, \lfloor x \rceil + \delta\sigma]} z_i g_i \tag{7}$$

## 2.3 RELATION TO STOCHASTIC ROUNDING

One of the pioneering works in neural network quantization has been the work of Gupta et al. (2015); it introduced stochastic rounding, a technique that is one of the most popular approaches for training neural networks with reduced numerical precision. Instead of rounding to the nearest representable value, the stochastic rounding procedure selects one of the two closest grid points with probability depending on the distance of the high precision input from these grid points. In fact, we can view stochastic rounding as a special case of RQ where $p(\tilde{x}) = U(x - \frac{\alpha}{2}, x + \frac{\alpha}{2})$. This uniform distribution centered at $x$ of width equal to the grid width $\alpha$ generally has support only for the closest grid point. Discretizing this distribution to a categorical over the quantization grid however assigns probabilities to the two closest grid points as in stochastic rounding, following Equation 2:

$$p(\hat{x} = \left\lfloor \frac{x}{\alpha} \right\rfloor \alpha \,|\, x) = P(\tilde{x} \le (\left\lfloor \frac{x}{\alpha} \right\rfloor \alpha + \alpha/2)) - P(\tilde{x} < (\left\lfloor \frac{x}{\alpha} \right\rfloor \alpha - \alpha/2)) = \left\lceil \frac{x}{\alpha} \right\rceil - \frac{x}{\alpha}. \tag{8}$$

Stochastic rounding has proven to be a very powerful quantization scheme, even though it relies on biased gradient estimates for the rounding procedure. On the one hand, RQ provides a way to circumvent this estimator at the cost of optimizing a surrogate objective. On the other hand, RQ ST makes use of the unreasonably effective straight-through estimator as used in Jang et al. (2016)

to avoid optimizing a surrogate objective, at the cost of biased gradients. Compared to stochastic rounding, RQ ST further allows sampling of not only the two closest grid points, but also has support for more distant ones depending on the estimated input noise $\sigma$. Intuitively, this allows for larger steps in the input space without first having to decrease variance at the traversion between grid sections.

## 3  RELATED WORK

In this work we focus on hardware oriented quantization approaches. As opposed to methods that focus only on weight quantization and network compression for a reduced memory footprint, quantizing all operations within the network aims to additionally provide reduced execution times. Within the body of work that considers quantizing weights and activations fall papers using stochastic rounding (Gupta et al., 2015; Hubara et al., 2016; Gysel et al., 2018; Wu et al., 2018). (Wu et al., 2018) also consider quantized backpropagation, which is out-of-scope for this work.

Furthermore, another line of work considers binarizing (Courbariaux et al., 2015; Zhou et al., 2018) or ternarizing (Li et al., 2016; Zhou et al., 2018) weights and activations (Hubara et al., 2016; Rastegari et al., 2016; Zhou et al., 2016) via the straight-through gradient estimator (Bengio et al., 2013); these allow for fast implementations of convolutions using only bit-shift operations. In a similar vein, the straight through estimator has also been used in Cai et al. (2017); Faraone et al. (2018); Jacob et al. (2017); Zhou et al. (2017); Mishra & Marr (2017) for quantizing neural networks to arbitrary bit-precision. In these approaches, the full precision weights that are updated during training correspond to the means of the logistic distributions that are used in RQ. Furthermore, Jacob et al. (2017) maintains moving averages for the minimum and maximum observed values for activations while parameterises the network's weights' grids via their minimum and maximum values directly. This fixed-point grid is therefore learned during training, however without gradient descent; unlike the proposed RQ. Alternatively, instead of discretizing real valued weights, Shayer et al. (2018) directly optimize discrete distributions over them. While providing promising results, this approach does not generalize straightforwardly to activation quantization. A bayesian approach to binarized models was taken in Soudry et al. (2014), which provided encouraging results on small scale experiments with an ensemble of quantized models sampled from the approximate posterior distribution. For small vocabulary sizes (e.g. ternary weights / activations) Yin et al. (2016) proposed explicit formulas to compute the closest (according to the Euclidean distance) quantized value.

Another line of work quantizes networks through regularization. (Louizos et al., 2017a) formulate a variational approach that allows for heuristically determining the required bit-width precision for each weight of the model. Improving upon this work, (Achterhold et al., 2018) proposed a quantizing prior that encourages ternary weights during training. Similarly to RQ, this method also allows for optimizing the scale of the ternary grid. In contrast to RQ, this is only done implicitly via the regularization term. One drawback of these approaches is that the strength of the regularization decays with the amount of training data, thus potentially reducing their effectiveness on large datasets. Alternatively, one could directly regularize towards a set of specific values via the approach described at Yin et al. (2018).

Weights in a neural network are usually not distributed uniformly within a layer. As a result, performing non-uniform quantization is usually more effective. (Baskin et al., 2018) employ a stochastic quantizer by first uniformizing the weight or activation distribution through a non-linear transformation and then injecting uniform noise into this transformed space. (Polino et al., 2018) propose a version of their method in which the quantizer's code book is learned by gradient descent, resulting in a non-uniformly spaced grid. Another line of works quantizes by clustering and therefore falls into this category; (Han et al., 2015; Ullrich et al., 2017) represent each of the weights by the centroid of its closest cluster. While such non-uniform techniques can be indeed effective, they do not allow for efficient implementations on todays hardware. Nevertheless, there is encouraging recent work (Zhang et al., 2018) on non-uniform grids that can be implemented with bit operations.

Within the litereterature on quantizing neural networks there are many approaches that are orthogonal to our work and could potentially be combined for additional improvements. (Mishra & Marr, 2017; Polino et al., 2018) use knowledge distillation techniques to good effect, whereas works such as (Mishra et al., 2017) modify the architecture to compensate for lower precision computations. (Zhou et al., 2017; 2018; Baskin et al., 2018) perform quantization in an step-by-step manner going from input layer to output, thus allowing the later layers to more easily adapt to the rounding errors

introduced. Polino et al. (2018); Faraone et al. (2018) further employ "bucketing", where small groups of weights share a grid, instead of one grid per layer. As an example from Polino et al. (2018), a bucket size of 256 weights per grid on Resnet-18 translates to $\sim 91.4k$ separate scaling factors / offsets as opposed to 22 in RQ.

# 4 EXPERIMENTS

For the subsequent experiments RQ will correspond to the proposed procedure that has concrete sampling and RQ ST will correspond to the proposed procedure that uses the Gumbel-softmax straight-through estimator (Jang et al., 2016) for the gradient. We did not optimize an offset for the grids in order to be able to represent the number zero exactly, which allows for sparsity and is required for zero-padding. Furthermore we assumed a grid that starts from zero when quantizing the outputs of ReLU. We provide further details on the experimental settings at Appendix A. We will also provide results of our own implementation of stochastic rounding (Gupta et al., 2015) with the dynamic fixed point format (Gysel et al., 2018) (SR+DR). Here we used the same hyperparameters as for RQ. All experiments were implemented with TensorFlow (Abadi et al., 2015), using the Keras library (Chollet et al., 2015).

## 4.1 LeNet-5 ON MNIST AND VGG7 ON CIFAR 10

For the first task we considered the toy LeNet-5 network trained on MNIST with the 32C5 - MP2 - 64C5 - MP2 - 512FC - Softmax architecture and the VGG 2x(128C3) - MP2 - 2x(256C3) - MP2 - 2x(512C3) - MP2 - 1024FC - Softmax architecture on the CIFAR 10 dataset. Details about the hyperparameter settings can be found in Appendix A.

By observing the results in Table 1, we see that our method can achieve competitive results that improve upon several recent works on neural network quantization. Considering that we achieve lower test error for 8 bit quantization than the high-precision models, we can see how RQ has a regularizing effect. Generally speaking we found that the gradient variance for low bit-widths (i.e. 2-4 bits) in RQ needs to be kept in check through appropriate learning rates.

## 4.2 RESNET-18 AND MOBILENET ON IMAGENET

In order to demonstrate the effectiveness of our proposed approach on large scale tasks we considered the task of quantizing a Resnet-18 (He et al., 2016) as well as a Mobilenet (Howard et al., 2017) trained on the Imagenet (ILSVRC2012) dataset. For the Resnet-18 experiment, we started from a pre-trained full precision model that was trained for 90 epochs. We provide further details about the training procedure in Appendix C. The Mobilenet was initialized with the pretrained model available on the tensorflow github repository[1]. We quantized the weights of all layers, post ReLU activations and average pooling layer for various bit-widths via fine-tuning for ten epochs. Further details can be found in Appendix C.

Some of the existing quantization works do not quantize the first (and sometimes) last layer. Doing so simplifies the problem but it can, depending on the model and input dimensions, significantly increase the amount of computation required. We therefore make use of the bit operations (BOPs) metric (Baskin et al., 2018), which can be seen as a proxy for the execution speed on appropriate hardware. In BOPs, the impact of not quantizing the first layer in, for example, the Resnet-18 model on Imagenet, becomes apparent: keeping the first layer in full precision requires roughly 1.3 times as many BOPs for one forward pass through the whole network compared to quantizing all weights and activations to 5 bits.

Figure 4 compares a wide range of methods in terms of accuracy and BOPs. We choose to compare only against methods that employ fixed-point quantization on Resnet-18 and Mobilenet, hence do not compare with non-uniform quantization techniques, such as the one described at Baskin et al. (2018). In addition to our own implementation of (Gupta et al., 2015) with the dynamic fixed point format (Gysel et al., 2018), we also report results of "rounding". This corresponds to simply rounding the pre-trained high-precision model followed by re-estimation of the batchnorm statistics. The grid

---

[1]`https://github.com/tensorflow/models/blob/master/research/slim/nets/mobilenet_v1.md`

Table 1: Test error (%) on MNIST and CIFAR 10 using LeNet5-Caffe and VGG-7 respectively. Two and four bit for VGG with SR+DR resulted in a big gap between training and validation accuracy, so we omit those results.

| Method | # Bits weights/act. | MNIST | CIFAR 10 |
|---|---|---|---|
| Original | 32/32 | 0.64 | 6.95 |
| SR+DR | 8/8 | 0.58 | 7.06 |
| (Gupta et al., 2015; Gysel et al., 2018) | 4/4 | 0.66 | - |
| | 2/2 | 1.03 | - |
| Deep Comp. (Han et al., 2015) | (5-8)/32 | 0.74 | - |
| TWN (Li et al., 2016) | 2/32 | 0.65[a] | 7.44 |
| BWN (Rastegari et al., 2016) | 1/32 | - | 9.88 |
| XNOR-net (Rastegari et al., 2016) | 1/1 | - | 10.17 |
| SWS (Ullrich et al., 2017) | 3/32 | 0.97 | - |
| Bayesian Comp. (Louizos et al., 2017a) | (7-18)/32 | 1.00 | - |
| VNQ (Achterhold et al., 2018) | 2/32 | 0.73 | - |
| WAGE (Wu et al., 2018) | 2/8 | 0.40 | 6.78 |
| LR Net (Shayer et al., 2018)[b] | 1/32 | 0.53[a] | 6.82 |
| | 2/32 | 0.50[a] | 6.74 |
| RQ (ours) | 8/8 | 0.55 | 6.70 |
| | 4/4 | 0.58 | 8.43 |
| | 2/2 | 0.76 | 11.75 |
| RQ ST (ours) | 8/8 | 0.56 | 6.72 |
| | 4/4 | 0.61 | 7.96 |
| | 2/2 | 0.63 | 9.08 |

[a]With batch normalization after convolution
[b]Last layer in full precision

in this case is defined as the initial grid used for fine-tuning with RQ. For batchnorm re-estimation and grid initialization, please confer Appendix A.

In Figure 4a we observe that on ResNet-18 the RQ variants form the "Pareto frontier" in the trade-off between accuracy and efficiency, along with SYQ, Apprentice and Jacob et al. (2017). SYQ, however, employs "bucketing" and Apprentice uses distillation, both of which can be combined with RQ and improve performance. Jacob et al. (2017) does better than RQ with 8 bits, however RQ improved w.r.t. to its pretrained model, whereas Jacob et al. (2017) decreased slightly. For experimental details with Jacob et al. (2017), please confer Appendix C.1. SR+DR underperforms in this setting and is worse than simple rounding for 5 to 8 bits.

For Mobilenet, 4b shows that RQ is competitive to existing approaches. Simple rounding resulted in almost random chance for all of the bit configurations. SR+DR shows its strength for the 8 bit scenario, while in the lower bit regime, RQ outperforms competitive approaches.

## 5 DISCUSSION

We have introduced Relaxed Quantization (RQ), a powerful and versatile algorithm for learning low-bit neural networks using a uniform quantization scheme. As such, the models trained by this method can be easily transferred and executed on low-bit fixed point chipsets. We have extensively evaluated RQ on various image classification benchmarks and have shown that it allows for the better trade-offs between accuracy and bit operations per second.

Future hardware might enable us to cheaply do non-uniform quantization, for which this method can be easily extended. (Lai et al., 2017; Ortiz et al., 2018) for example, show the benefits of low-bit floating point weights that can be efficiently implemented in hardware. The floating point quantization grid can be easily learned with RQ by redefining $\hat{\mathcal{G}}$. General non-uniform quantization, as described

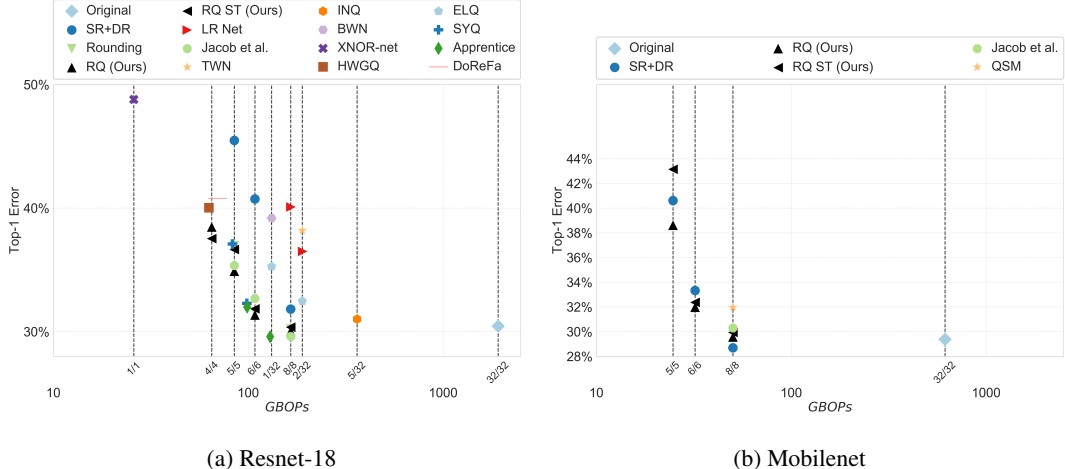

(a) Resnet-18               (b) Mobilenet

Figure 4: Best viewed in color. Comparison of various methods on Resnet-18 and Mobilenet according to top-1 error (on the y-axis) and bit operations (on the x-axis) computed according to the formula described in Baskin et al. (2018). Each dashed line corresponds to employing a specific bit configuration for every layer's weights and activations. Values for top-1 and top-5 errors are given in Table 2 in the Appendix. We compare against multiple works that employ fixed-point quantization: SR+DR (Gupta et al., 2015; Gysel et al., 2018), LR Net (Shayer et al., 2018), Jacob et al. (2017), TWN (Li et al., 2016), INQ (Zhou et al., 2017), BWN (Rastegari et al., 2016), XNOR-net (Rastegari et al., 2016), DoReFa (Zhou et al., 2016), HWGQ (Cai et al., 2017), ELQ Zhou et al. (2018), SYQ (Faraone et al., 2018), Apprentice (Mishra & Marr, 2017), QSM (Sheng et al., 2018) and rounding.

for example in (Baskin et al., 2018), is a natural extension to RQ, whose exploration we leave to future work. For example, we could experiment with a base grid that is defined as in Zhang et al. (2018). Currently, the bit-width of every quantizer is determined beforehand, but in future work we will explore learning the required bit precision within this framework. In our experiments, batch normalization was implemented as a sequence of convolution, batch normalization and quantization. On a low-precision chip, however, batch normalization would be "folded" (Jacob et al., 2017) into the kernel and bias of the convolution, the result of which is then rounded to low precision. In order to accurately reflect this folding at test time, future work on the proposed algorithm will emulate folded batchnorm at training time and learn the corresponding quantization grid of the modified kernel and bias. For fast model evaluation on low-precision hardware, quantization goes hand-in-hand with network pruning. The proposed method is orthogonal to pruning methods such as, for example, $L_0$ regularization (Louizos et al., 2017b), which allows for group sparsity and pruning of hidden units.

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

## A    EXPERIMENTAL DETAILS

The grid width $\alpha$ of each grid was initialized according to the bit-width $b$ and the maximum and minimum values of the input $x$ to the quantizer[2]. Since the inputs $\tilde{x}$ in both cases for our approach are stochastic it makes sense to assume a width for the grid that is slightly larger than the standard width $t = (\max(x) - \min(x))/2^b$; for the activations, whenever $b > 4$, we initialize $\alpha = t + 3t/2^b$, for $4 \geq b > 2$ we used $\alpha = t + 3t/2^{b+1}$ and finally for $b = 2$ we used $\alpha = t$. Since with ReLU activations the magnitude can become quite large (thus leading to increased quantization noise for smaller bit widths), this scheme keeps the noise injected to the network in check. For the weights we always used an initial $\alpha = t + 3t/2^b$. The standard deviation of the logistic noise $\sigma$ was initialized to be three times smaller than the width $\alpha$, i.e. $\sigma = \alpha/3$. Under this specification, most of the probability mass of the logistic distribution is initially (roughly) in the bins containing the closest grid point and its' two neighbors.

The moving averages of layer statistics that are aggregated during the training phase for the batch normalization do not necessarily reflect the statistics of the quantized model accurately. Even though RQ aims to minimize the gap between training and testing phase, we found that the aggregated statistics in combination with the learned scale and shift parameters of batch normalization lead to decreased test performance. In order to avoid this drop in accuracy, we apply the insights from (Peters & Welling, 2018) and recompute the statistics of the quantized model before reporting the final test error rate. The final models were determined through early stopping using the validation loss computed with minibatch statistics, in case the model uses batch normalization.

For the MNIST experiment we rescaled the input to the [-1, 1] range, employed no regularization and the network was trained with Adam (Kingma & Ba, 2014) and a batch size of 128. We used a local grid whenever the bit width was larger than 2 for both, weights and biases (shared grid parameters), as well as for the ouputs of the ReLU, with $\delta = 3$. For the 8 and 4 bit networks we used a temperature $\lambda$ of 2 whereas for the 2 bit models we used a temperature of 1 for RQ. We trained the 8 and 4 bit networks for 100 epochs using a learning rate of 1e-3 and the 2 bit networks for 200 epochs with a learning rate of 5e-4. In all of the cases the learning rate was annealed to zero during the last 50 epochs.

For the CIFAR 10 experiment, the hyperparameters were chosen identically to the LeNet-5 experiments except a few differences. We chose a learning rate ot 1e-4 instead of 1e-3 for 8 and 4 bit networks and trained for 300 epochs with a batch size of 100. We also included a weight decay term of 1e-4 for the 8 bit networks. For the 2 bit model we started with a learning rate of 1e-3. The VGG model contains a batch normalization layer after every convolutional layer, but preceeded by max pooling, if present.

## B    CONVERGENCE SPEED OF VGG ON CIFAR 10

Training a neural network with RQ imposes an additional sampling burden for every weight and activation in the network. Here, we investigate whether the extra "noise" that is introduced hampers the convergence speed of the network when we train from a random initialization. We recorded the learning curves for a 2/2 bit RQ-VGG network on CIFAR 10 (as this quantization level exhibits the largest amount of noise) and compare it to the full precision baseline. The results can be seen in

---

[2]For activations we computed the minimum and maximum on a random minibatch of inputs.

Figure 5. As we can observe, the 2/2 bit network has qualitatively similar trends to the full precision baseline. Therefore we can conclude that the noise is not detrimental for the task at hand, at least for this particular model. In terms of wall-clock time, training the RQ model with a full (4 elements) grid took approximately 15 times as long as the high-precision baseline with an implementation in Tensorflow v1.11.0 and running on a single Titan-X Nvidia GPU.

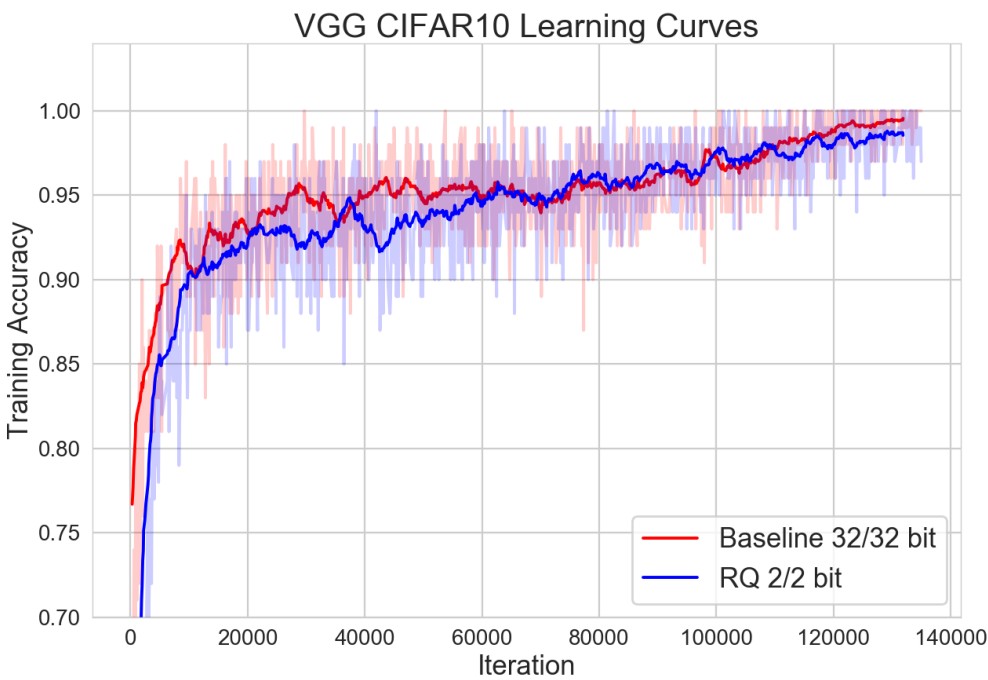

Figure 5: Learning curves for the VGG on CIFAR 10.

## C    IMAGENET DETAILS

Each channel of the input images was preprocessed by subtracting the mean and dividing by the standard deviation of that channel across the training set. We then resized the images such that the shorter side is set to 256 and then applied random 224x224 crops and random horizontal flips for data augmentation. For evaluation we consider the center 224x224 crop of the images.

We trained the base Resnet-18 model with stochastic gradient descent, a batch size of 128, nesterov momentum of 0.9 and a learning rate of 0.1 which was multiplied by 0.1 at the 30th and 60th epoch. We also applied weight decay with a strength of 1e-4. For the quantized model fine-tuning phase, we used Adam with a learning rate of $5e^{-6}$, a batch size of 24 and a momentum of 0.99. We used a temperature of 2 for both RQ variants. Following the strategy in (Jacob et al., 2017), we did not quantize the biases.

Table 2 contains the error rates for Resnet-18 and Mobilenet on which Figure 1 is based on. Algorithm and architecture specific changes are mentioned explicitly through footnotes.

### C.1    JACOB ET AL. (2017) FOR RESNET18

We used the code provided at `https://github.com/tensorflow/models/tree/master/official/resnet` and modified the construction of the training and evaluation graph by inserting quantization operations provided by the `tensorflow.contrib.quantize` package. In a first step, the unmodified code was used to train a high-precision Resnet18 model using the hyper-parameter settings for the learning rate scheduling that are provided in the github repository. More specifically, the model was trained for 90 epochs with a batch size of 128. The learning rate scheduling involved a "warm up" period in which the learning rate was annealed from zero to 0.64

Table 2: Top-1 and top-5 error (%) with Resnet18 and Mobilenet (full resolution and multiplier of one) on Imagenet

| Method | # Bits weights/act. | Resnet18 Top-1 | Resnet18 Top-5 | Mobilenet Top-1 | Mobilenet Top-5 |
|---|---|---|---|---|---|
| Original | 32/32 | 30.46 | 10.81 | 29.39 | 10.53 |
| SR+DR | 8/8 | 31.83 | 11.48 | 28.70 | 10.04 |
| (Gupta et al., 2015; Gysel et al., 2018) | 6/6 | 40.75 | 16.90 | 33.34 | 12.83 |
| | 5/5 | 45.48 | 20.16 | 40.61 | 17.65 |
| Rounding | 8/8 | 30.22 | 10.60 | - | - |
| | 6/6 | 31.61 | 11.32 | - | - |
| | 5/5 | 36.97 | 14.95 | - | - |
| | 4/4 | 78.79 | 57.10 | - | - |
| (Jacob et al., 2017)[a] | 8/8 | 29.62 | 10.45 | 30.30 | 10.50 |
| | 6/6 | 32.69 | 12.46 | - | - |
| | 5/5 | 35.36 | 13.33 | - | - |
| LR Net (Shayer et al., 2018) | 1/32[b] | 40.10 | 17.70 | - | - |
| | 2/32[c] | 36.50 | 15.20 | - | - |
| QSM (Sheng et al., 2018)[a d] | 8/8 | - | - | 31.97 | - |
| TWN (Li et al., 2016) | 2/32 | 38.20 | 15.80 | - | - |
| INQ (Zhou et al., 2017) | 5/32 | 31.02 | 10.90 | - | - |
| BWN (Rastegari et al., 2016) | 1/32 | 39.20 | 17.00 | - | - |
| XNOR-net (Rastegari et al., 2016) | 1/1 | 48.80 | 26.80 | - | - |
| HWGQ (Cai et al., 2017)[b] | 1/2 | 40.4 | 17.8 | - | - |
| DoReFa (Zhou et al., 2016)[be] | 1/4 | 40.8 | 18.5 | - | - |
| ELQ (Zhou et al., 2018) | 1/32 | 35.28 | 13.96 | - | - |
| | 2/32 | 32.48 | 11.95 | - | - |
| SYQ (Faraone et al., 2018)[f] | 1/8 | 37.1 | 15.4 | - | - |
| | 2/8 | 32.3 | 12.2 | - | - |
| Apprentice (Mishra & Marr, 2017)[b] | 2/8 | 32 | — | — | — |
| | 4/8 | 29.6 | — | — | — |
| RQ (ours) | 8/8 | 30.03 | 10.56 | 29.57 | 10.58 |
| | 6/6 | 31.35 | 11.22 | 31.98 | 12.00 |
| | 5/5 | 34.90 | 13.43 | 38.62 | 16.27 |
| | 4/4 | 38.48 | 16.01 | - | - |
| RQ ST (ours) | 8/8 | 30.37 | 10.67 | 29.94 | 10.48 |
| | 6/6 | 31.85 | 11.62 | 32.38 | 12.22 |
| | 5/5 | 36.65 | 14.54 | 43.15 | 19.65 |
| | 4/4 | 37.54 | 15.22 | - | - |

[a]Includes folded batch normalization

[b]First and last layer not quantized

[c]First layer not quantized

[d]Modified architecture

[e]Results taken from https://github.com/tensorpack/tensorpack/blob/master/examples/DoReFa-Net/resnet-dorefa.py

[f]Weights of first and last layer not quantized

over the first $50k$ steps, after which it was divided by 10 after epochs $30, 60$ and $80$ respectively. Gradients were modified using a momentum of $0.9$. Final test performance under this procedure is $29.53\%$ top-1 error and $10.44\%$ top-5 error. From the high-precision model checkpoint, the final quantized model was then fine-tuned for 10 epochs using a constant learning rate of $1e^{-4}$ and mo-

mentum of $0.9$. We did not freeze the moving averages of the batch normalization layers. Finally, we found that re-estimating the batchnorm statistics was harmful for this algorithm. We hypothesise that this is due to the usage of folded batch normalization, which incorporates the statistics into the construction of the grid at training time.

## C.2 JACOB ET AL. (2017) FOR MOBILENET

The $8/8$ bit results for quantizing Mobilenet provided in table 2 are read off from Figure 4.1 in Jacob et al. (2017). The pre-trained models published at `https://github.com/tensorflow/models/blob/master/research/slim/nets/mobilenet_v1.md` originally reflected that number up until commit `4415c2613b0c74032a7c631769ef9fa7f5477d88`, but have since been updated to improved error rates of 29.9 and 11.1 respectively. Unfortunately, there are several conflicting sources for quantized Mobilenet results and pretrained-models within the tensorflow github repository. `https://github.com/tensorflow/tensorflow/blob/master/tensorflow/contrib/lite/g3doc/models.md#image-classification-quantized-models`, for example, reports error rates of 30.0 and 11.0, whereas at `https://github.com/tensorflow/tensorflow/tree/master/tensorflow/contrib/quantize` the reported top-1 error rate is 30.3.

We attempted to use the provided training scripts in the `https://github.com/tensorflow/models/blob/master/research/slim` repository to train lower-bit mobilenet variants, but did not succeed in doing so. We experimented with learning rates in the range of $[5e^{-6}, 5e^{-5}, 1e^{-4}]$ for 5/5, 6/6 and 8/8 bit-width variants, but could not achieve significant accuracy improvements within the first 10 epochs of fine-tuning of the high-precision model published at `https://github.com/tensorflow/models/blob/master/research/slim/nets/mobilenet_v1.md`. After 10 epochs, the $8/8$ version achieved 31.39 top-1 error with a learning rate of $1e^{-4}$ and as such is worse than the published results. We therefore chose to only include the published numbers for the $8/8$ bit model and leave addition hyperparameter tuning to future work.

