# OpenReview forum: "Relaxed Quantization for Discretized Neural Networks"
_ICLR.cc/2019/Conference_

### Official Review · AnonReviewer2 · 2018-11-01
**Fairly straight-forward ideas but good results and solid empirical work**

**Rating:** 7
**Confidence:** 4

**Review:**

Summary
=======
This paper introduces a method for learning neural networks with quantized weights and activations. The main idea is to stochastically – rather than deterministically – quantize values, and to replace the resulting categorical distribution over quantized values with a continuous relaxation (the "concrete distribution" or "Gumbel-Softax distribution"; Maddison et al., 2016; Jang et al., 2016). Good empirical performance is demonstrated for LeNet-5 applied to MNIST, VGG applied to CIFAR-10, and MobileNet and ResNet-18 applied to ImageNet.

Review
======
Relevance:
Training non-differentiable neural networks is a challenging and important problem for several applications and a frequent topic at ICLR.

Novelty:
Conceptually, the proposed approach seems like a straight-forward application/extension of existing methods, but I'm unaware of any paper which uses the concrete distribution for the express purpose of improved efficiency as in this paper. There is a thorough discussion of related work, although I was missing Williams (1992), who used stochastic rounding before Gupta et al. (2015), and Soudry et al. (2014), who introduced a Bayesian approach to deal with discrete weights and activations.

Results:
The empirical work is thorough, achieving state-of-the-art results in several classification benchmarks. It would be interesting to see how well these methods perform in other tasks (e.g., compression or even regression), even though the literature on quantization seems to focus on classification.

Clarity:
The paper is well written and clear.

---

> ### Author Response · Authors · 2018-11-12
> **We will update the related work section and different tasks are left for future work.**
>
> Dear Reviewer 2,
>
> Thank you for your review and comments for approval.
>
> We will make sure to update the related work section with the work of Soudry et al. (2014). As for Williams (1992); to our understanding the focus of that paper was to introduce the unbiased score function estimator REINFORCE for the gradient of an expectation of a non-differentiable function. In this sense, Williams (1992) is more of a related work to the concrete / Gumbel-softmax approaches, rather than the stochastic rounding of Gupta et al. (2015). We will update the submission to include a brief discussion between the REINFORCE and concrete / Gumbel-softmax as choices for the fourth element of RQ.
>
> Regarding experiments on different tasks; we agree that it would be interesting to check performance on tasks that require more “precision”, such as regression. We chose classification for this submission, as this provides a large amount of literature to compare against, and leave the exploration of different tasks for future work.

---

### Official Review · AnonReviewer1 · 2018-11-03
**New approach to quantizing activations, SotA/competitive on several real image problems**

**Rating:** 7
**Confidence:** 3

**Review:**

Quality:
The work is well done. Experiments cover a range of problems and a range of quantization resolutions. Related work section in, particular, I thought was very nicely done. Empirical results are strong.

In section 2.2, it bothers me that the amount of bias introduced by using the local grid approximation is never really assessed. How much probability mass is left out by truncating the Gumbel-softmax, in practice?

Clarity:
Well presented. I believe I'd be able to implement this, as a practitioner.

Originality:
Nice to see the concrete approximation having an impact in the quantization space.

Significance:
Quantization has obvious practical interest. The regularization aspect is striking (quantization yielded slightly improved test error on CIFAR-10; is that w/in the error bars?). A recent work [https://arxiv.org/abs/1804.05862] links model compressibility to generalization; while this work is more focused on activations, there is no reason that it couldn't be used for weights as well.

Nits:
top of pg 6 'reduced execution speeds' -> times, or increased exec speeds
'sparcity' misspelled

---

> ### Author Response · Authors · 2018-11-12
> **Only 2% of the probability mass is truncated and the regularization aspect is definitely interesting.**
>
> Dear Reviewer 1,
>
> Thank you for your review and comments for approval.
>
> Regarding the bias of the local grid approximation; we mentioned in the main text that the local grid is constructed such that points that are within \delta standard deviations from the mean are always part of it. For all of our experiments, we set \delta = 3, which means that, roughly, only 2% of the probability mass of the logistic distribution is truncated. Unfortunately, due to lack of space we moved these experimental details about hyperparameters in the appendix.
>
> Regarding the regularization aspect; indeed we observed that for VGG, quantizing to 8/8 bits resulted in consistent improved test errors. We are definitely aware of [https://arxiv.org/abs/1804.05862] and believe that further research in this direction is a fruitful direction.

---

### Official Review · AnonReviewer3 · 2018-11-07
**Good paper that proposes an effective method to train neural networks with quantized reduced precision synapses and activations**

**Rating:** 7
**Confidence:** 4

**Review:**

The authors proposes a unified and general way of training neural network with reduced precision quantized synaptic weights and activations. The use case where such a quantization can be of use is the deployment of neural network models on resource constrained devices, such as mobile phones and embedded devices.

The paper is very well organized and systematically illustrates and motivates the ingredients that allows the authors to achieve their goal: a quantization grid with learnable position and range, stochastic quantization due to noise, and relaxing the hard categorical quantization assignment to a concrete distribution.
The authors then validate their method on several architectures (LeNet-5, VGG7, Resnet and mobilnet) on several datasets (MNIST, CIFAR10 and ImageNet) demonstrating competitive results both in terms of precision reduction and accuracy.

Minor comments:
- It would be interesting to know whether training with the proposed relaxed quantization method is slower than with full-precision activations and weights. It would have been informative to show learning curves comparing learning speed in the two cases.
- It seems that this work could be generalized in a relatively straight-forward way to a case in which the quantization grid is not uniform, but instead all quantization interval are being optimized independently. It would have been interesting if the authors discussed this scenario, or at least motivated why they only considered quantization on a regular grid.

---

> ### Author Response · Authors · 2018-11-12
> **There is extra overhead involved, we will add exemplary learning curves and non-uniform grids are left for future work.**
>
> Dear Reviewer 3,
>
> Thank you for your review and comments for approval.
>
> Addressing the first point of training speed: training a neural network with the proposed method indeed imposes an additional burden in computing and sampling the categorical probabilities over the local grid for every weight and activation in the network. As such, this method introduces an overhead which is not present in methods that rely on deterministic rounding and the straight-through estimator for gradients. As for convergence speed, we will include an exemplary learning curve for the 32/32, 8/8 and 2/2 bit VGG in the appendix.
>
> Addressing your second point about non-uniform grids: as you have stated, this method can be easily extended to non-uniform grids. Doing so would only require evaluating the CDF of the continuous signal at different points on the real line. We have mentioned this possibility of non-uniform grids in the conclusion to our work. The reason for why we consider uniform grids only lies in that non-uniform grids, although more powerful, generally do not allow for a  straightforward implementation in today’s low-bit hardware. We mention that we explicitly focus on uniform grids for this specific reason of hardware suitability.

---

### Public Comment · (anonymous) · 2018-11-08
**Isn't this a duplicated submission as DARTS?**

Isn't this a duplicated submission as DARTS?
https://openreview.net/forum?id=S1eYHoC5FX

---

> ### Author Response · Authors · 2018-11-12
> **This submission is by no means a duplicate.**
>
> Dear anonymous commenter,
>
> Although the proposed relaxed quantization method shares some similarities with DARTS, this submission is by no means a duplicate. The similarities can be summarized in that both methods consider the computation of gradients through a non-differentiable selection mechanism. In our work, selection happens between grid-points. In DARTS, selection happens between choices of neural network architecture elements. Please note that in our work, we propose to use the relaxation of the categorical choice in order to draw samples, whereas in DARTS, the relaxation is performed by learning a weighted average.
>
> We hope to have interpreted and answered your question appropriately. Please let us know if there are any remaining questions.

---

### Public Comment · (anonymous) · 2018-11-12
**Very poor performance on ImageNet.**

Dear authors and reviewers,

Please check the performance of the current state-of-the-art approaches [1, 2, 3] on ImageNet.  For 4-bit Resnet-18, they can achieve near lossless results. For example, in LQ-Net [1], it only has 0.3% and 0.4% Top1 and Top5 accuracy drop, respectively. But in this paper, it has more than 7% Top-1 accuracy drop.  Even uniform quantization approach DOREFA-Net performs much better than this submission.
And I don't know why this submission just "ignores" these approaches?

References (Only list three of them) :
[1]: "LQ-Nets: Learned Quantization for Highly Accurate and Compact Deep Neural Networks". ECCV2018.
[2]: "PACT: Parameterized Clipping Activation for Quantized Neural Networks". https://arxiv.org/pdf/1805.06085
[3]: "DoReFa-Net: Training Low Bitwidth Convolutional Neural Networks with Low Bitwidth Gradients",
 https://arxiv.org/abs/1606.06160.

---

> ### Author Response · Authors · 2018-11-12
> **We beg to differ on the very poor performance on Imagenet, there are important details that have to be taken into account.**
>
> Dear anonymous commenter,
>
> Thank you for the interest in our work and for bringing [1, 2] into our attention. First of all, we would like to respectfully disagree with the comment of “very poor performance on Imagenet”. More specifically, we believe that there some important differences between e.g. [1] and this work that do not lend to a fair comparison.
>
> To further elaborate, in [1] the authors propose a non-uniform quantization grid while arguing for it being compatible with bit operations. In our work we focus on uniform quantization grids because they lend themselves to straight-forward implementation on current hardware. The more powerful grid proposed in [1] is orthogonal to the contributions of this work and can be further employed to boost the performance of RQ. We will update the paper with an appropriate discussion.
>
> It is also worth pointing out several subtleties w.r.t. the hyperparameters and details of the experiments in [1], that make a fair comparison difficult:
>
>  - First of all, it seems that [1] used a modified pre-activation ResNet18 architecture (judging from the paper and publicly available code of LQ-net), which is different from the standard ResNet18 architecture that we and the other baselines employed (our ResNet18 was based on https://github.com/fchollet/deep-learning-models/blob/master/resnet50.py).
>
>  - Secondly,  [1, 2, 3] did not quantize the first and last layer of the network; while this can allow for better performance in terms of top1-5 accuracy it also negatively affects the model efficiency, as the BOP count will be (much) higher than our 4/4 model. For example, on a ResNet-18 with 4/4 bits and no quantization of the first and last layer we get approximately 24 GBOPs extra (according to the metric we used in the submission) compared to an 8/8 bit model that quantizes all weights and activations. In this sense, the 8 bit RQ has better accuracy while also maintaining better efficiency than the 4 bit LQ-net. Similar arguments can be made for [2, 3].
>
>  - Thirdly, it seems that [1] also used a much more flexible quantization grid construction for the weights; it assumed a separate quantization grid per channel, rather than per an entire layer (as in this work). This further increases the flexibility of the quantization model but it does make hardware implementations more difficult and less efficient. Similarly as before, such a grid construction is easily applied to RQ and can similarly further improve performance.
>
> Finally, we did not compare against [3] as it did not provide any results for the architectures we compare against in this paper. Their imagenet results were obtained using a variant of the AlexNet architecture, whereas we compare on the more recent ResNet18 and MobileNet. After reading [1] however, we were made aware of the Resnet18 results presented in their git repo, so we will update the paper with those numbers. Similarly to [1, 2], not quantizing the first and last layer results into worse accuracy / efficiency trade-offs than RQ.

---

> > ### Public Comment · (anonymous) · 2018-11-13
> > **Response to authors**
> >
> > Dear authors,
> >
> > Thx for your detailed answer, but I still have some doubts.
> >
> > 1):  You have clarified differences between this submission and [1] via analysis. But I don't think it is too hard to implement your approach using pre-activation ResNet-18 and keep the first and the last layer to full-precision. Without any results provided, I am still not sure to what extend these modifications can influence the performance. After all, a 7% gap on ImageNet is not small.
> >
> > 2):  You argue that it is not hardware friendly to use a separate quantization grid per channel.  However, since you did not implement on any hardware device, your argument cannot convince me.  In fact, a NIPS2018 paper [1] this year claims that "heterogeneously binarized (mixed bitwise) systems yield FPGA- and ASIC-based implementations that are correspondingly more efficient in both circuit area and energy efficiency than their homogeneous counterparts."  In this paper, each parameter/activation has different bitwise, but they have shown that it is still efficient to implement on hardware platforms.
> >
> > And if you can provide any results here that will be better.  Thanks again for your patient answer.
> >
> > Reference:
> > [1]: "Heterogeneous Bitwidth Binarization in Convolutional Neural Networks", NIPS20`18.

---

> > > ### Author Response · Authors · 2018-11-13
> > > **BOP count metric normalizes some design choices, we targeted generally available chips on the market today**
> > >
> > > Dear anonymous commenter,
> > >
> > > Thank you for your additional comment. We hope to address your doubts adequately:
> > >
> > > 1) Indeed, implementing RQ for a pre-activation Resnet18 with the hyperparameters that you propose is feasible. Nevertheless, we also believe that it is not necessary: firstly, as we previously mentioned, the GBOP metric that we used in the submission “normalizes” against the choice of having a full precision first and last layer, therefore we can safely conclude that the 8/8 bit RQ model that quantizes everything is better, both BOP wise and accuracy wise, than the 4/4 bit LQ-Net model that does not quantize the first and last layer. Secondly, we chose to experiment with the standard ResNet18 architecture in order to be able to compare with the majority of the quantization literature. As a result, we do not believe that the experiments with the pre-activation ResNet18 will offer additional insight, besides allowing for a slightly more calibrated comparison against e.g LQ-Net or PACT. Instead, we believe that a completely different architecture (MobileNet) better complements our ResNet18 experiments.
> > >
> > > In summary, we hope to have convinced you of the practical importance of quantizing the first and last layers. On the side of experiments provided, we believe to have produced significant evidence in favour of RQ. The code to reproduce our results as well as to do additional experiments is currently undergoing regulatory approval. Please stay tuned for our announcement and feel free to contact us with questions about your own re-implementation once the contact details are available.
> > >
> > > 2) Thank you for the pointer to this work; we believe it provides interesting food for thought for future hardware choices. We base our argument not on a specific chipset, but argue for the properties of generally available chips on the market today. Examples of state-of-the-art chips that especially target fixed-point computations include: (Qualcomm) Hexacore 68X,  (Intel) movidius, (ARM) NEON. In case the application warrants specialized hardware (ASIC) or FPGAs, there will always be highly efficient specialized solutions that might allow for different bit-precisions (or even mix fixed-point and floating-point representations [1]. However it becomes increasingly difficult to find a fair basis of speed/accuracy comparison when allowing for arbitrary hardware implementations and to account for the additional overhead of e.g. channel-wise grids. Again, we believe our experimental efforts to lay sufficient claim for the validity of RQ by comparing many works that use fixed-point shared grids. Any additional modifications such as mixed-precision, channelwise-grids or any of the other strategies referenced in our paper are orthogonal to our method and it is reasonable to believe that including them will benefit RQ as well.
> > >
> > > [1] Deep Convolutional Neural Network Inference with Floating-point Weights and Fixed-point Activations: https://arxiv.org/pdf/1703.03073.pdf
> > >
> > > EDIT: After fixing the BOP count metric (see general comment), the 4/4 bit LQ-Net BOP count lies between the 5/5 and 6/6 bit RQ models. In this case we observe that the accuracy of RQ is slightly worse than the LQ-net for an approximately same count of BOPs, which could be explained due to the non-uniform grid and channel-wise kernel quantization.

---

> > > > ### Public Comment · (anonymous) · 2018-11-13
> > > > **Some questions about BOP count metric**
> > > >
> > > > Dear authors,
> > > >
> > > > Thanks for your answer and I learn a lot. But I find some (potential) mistakes in the BOP metric.
> > > >
> > > > 1): The width $w$ and height $h$ of the feature map are not included in the layer complexity. And I am sure this should be fixed.
> > > >
> > > > 2): Let us assume weights and activations are all binary (1-bit). Then the convolutional operations becomes XNOR and popcount, which are all bitwise operations.  So according to BOPs, the bitwise popcount complexity (for a single output calculation) is n{k^2}(2 + {\log _2}n{k^2}) . However, it doesn't make sense since this complexity holds for floating-point additions rather than bitwise operations.
> > > >
> > > > And could you check my claims?

---

> > > > > ### Author Response · Authors · 2018-11-14
> > > > > **Clarifications about BOP count**
> > > > >
> > > > > Dear anonymous commenter,
> > > > >
> > > > > Of course, you can find the answers below:
> > > > >
> > > > > 1) In our computation of a model’s BOP count, we do take height and width of the feature maps into account. The formula as stated in [1] is given for “a single output calculation” and correspondingly by us multiplied for a whole layer’s BOP count.
> > > > >
> > > > > 2) We merely aim to use the BOP count formula from [1] as a rough estimate of the actual BOPs of a given low-bit model, and not as an exact measure. Our aim was to have a sensible ranking of all the methods we compare. Indeed, for 1 bit weights and activations, the BOP approximation will be worse compared to fixed-point or floating-point networks.  We would like to point out that the formula came with its own set of assumptions, which are stated at [1]. We agree that the BOP count is not a perfect measure of model complexity or execution speed, however it does serve as a normalizer for the purpose of comparison. Finally we recognize that execution speed might be identical or higher for example for a 4/4 bit model on a chip with a dedicated 4/4 instruction set compared to a 3/3 bit model on the same chip, due to suboptimal kernels. A similar conclusion could be drawn for a chip that does not possess a fixed-point instruction set when comparing fixed-point to floating-point models. That is to say the final execution speed/accuracy trade-off is very dependent on the targeted hardware and any measure that tries to generalize across different chips will either be very complex or always remain approximative.
> > > > >
> > > > > [1] Baskin, C., Schwartz, E., Zheltonozhskii, E., Liss, N., Giryes, R., Bronstein, A. M., & Mendelson, A. (2018). UNIQ: Uniform Noise Injection for the Quantization of Neural Networks. arXiv preprint arXiv:1804.10969.

---

### Author Response · Authors · 2018-11-13
**Aggregation of feedback**

Dear reviewers and commenters,

We have updated the submission to include all of the discussed points, except for the learning curves for VGG as we are currently rerunning the experiments in order to track them. We will perform another update as soon as that is finished.

Please also note that we have updated Figure 4 that contains the Imagenet results. We updated our BOP count implementation to correctly take into account the 8bit input of the models that have a full precision first layer. This resulted in a lower BOP count for these models. Nevertheless we still observe that the RQ models lie on the Pareto frontier, hence the conclusions do not change.

EDIT: We have uploaded a new version of the paper that contains the VGG learning curves in the appendix.

---

### Meta-Review · Area_Chair1 · 2018-12-17
**new approach**

**Confidence:** 5
**Recommendation:** Accept (Poster)

**Metareview:**

This paper proposes an effective method to train neural networks with quantized reduced precision. It's fairly straight-forward idea and achieved good results and solid empirical work. reviewers have a consensus on acceptance.